# PROGRESSIVE VISUAL RELATIONSHIP INFERENCE

## ABSTRACT

As an important component of visual scene, visual relationship has received extensive attention in recent years. Most existing works directly utilize the rough visual appearance to represent visual relationships. Although they have been made tremendous progress, the study of visual relationship may be still far from perfect. This common idea may have three problems. 1) The similarity of space aggravates the ambiguity of predicate representation. 2) The differences between many visual relationships are subtle. 3) It lacks interpretability. To address these problems, we propose a novel method - **P**rogressive **V**isual **R**elationship **I**nference(**PVRI**) - which considers both rough visual appearance and fine-grained visual cues to gradually infer visual relationships. It includes the following three steps. 1) Known Cues Collection: firstly, we utilize Large Language Model(LLM) to collect the cues that may help infer visual relationships; 2) Unknown Cues Extraction: secondly, we design UCE strategy to extract the cues that are not defined by the text. 3) Progressive Inference: thirdly, we utilize the obtained cues to infer visual relationships. We demonstrate the effectiveness and efficiency of our method for the Visual Genome, Open Image V6 datasets.

## 1 INTRODUCTION

With the development of deep learning, people's interest in visual scene understanding has increased significantly in recent years(Ye & Xu (2024); Xin et al. (2024); Zhang et al. (2024b)). As an important component of visual scene, visual relationship has also received extensive attention such as: visual question answering(Qian et al. (2024); Lin et al. (2024); Gao et al. (2024)), visual relationship detection(Li et al. (2024b); Lu et al. (2016); Liang et al. (2018)) and scene graph generation(Li et al. (2024a); Zhao et al. (2024); Lin et al. (2024)). Most of them(Krishna et al. (2017); Tang et al. (2019); Chen et al. (2019); Zhang et al. (2019b)) directly utilize the rough visual appearance to represent visual relationship. Although they have been made tremendous progress, the study of visual relationship may be still far from perfect.

This common idea may have three problems. 1) **The similarity of space aggravates the ambiguity of predicate representation**. For example, "playing" may be similar to "on"(boy-playing-skateboard) or "near"(boy-playing-ball). We can always find a similar geometric relationship for any kind of visual relationship(as shown in fig. 1.a). The rough appearance representation is difficult to take into account their similarities and differences. 2) **The differences between some similar visual relationships are subtle**. For example, the difference between "stand on" and "walk on" is in the legs. The rough visual appearance is not enough to capture the nuances of these similar visual relationships. 3) **It lacks interpretability**. Visual relationship serves visual scene understanding, and its interpretability is directly related to the user's trust in the model. This idea is difficult to explain the differences between different predicates.

To address these problems, a primary challenge is **how to explain the similarities and differences of visual relationships**? A simple idea is to utilize **L**arge **L**anguage **M**odels(**LLM**, such as: GPT-4) to explain visual relationships through some fine-grained cues(as shown in fig. 1.b). Through these descriptions, we can well capture the subtle differences in similar visual relationships. For example, the difference between "stand on" and "walk on" on leg state. Followed by, a natural question that arises is: how to calculate the similarity of visual relationships? Some existing works(Zhang et al. (2024a); Zhou et al. (2020)) utilize the semantic knowledge to construct a hierarchy of predicates to reflect the similarity of predicates. Extensive experiments have proved that the semantic knowledge can effectively reflect the similarity between visual relationships. However, we recognize that it is

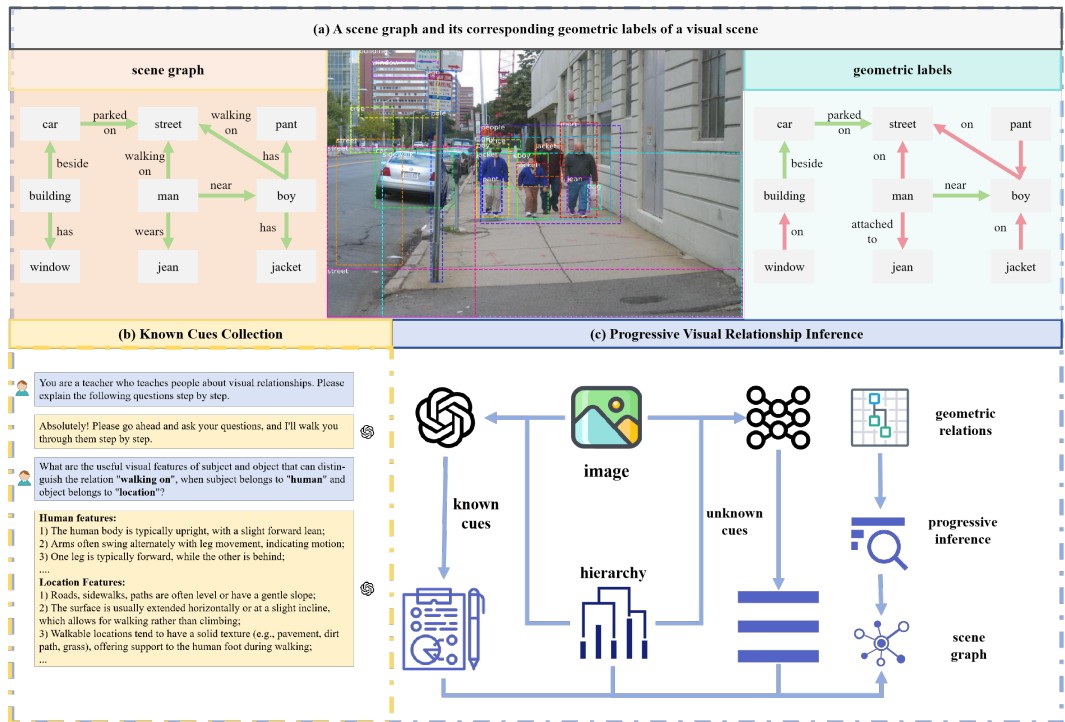

Figure 1: The overview of our idea. a) represents the scene graph of an image and its corresponding geometric labels, indicating that we can always find a similar geometric label for each visual relationship; b) represents the cue descriptions we obtained through LLM that helps to infer visual relationships, and is also a part of our KCC strategy; c) represents the process of our method.

difficult to explain the spatial similarity between predicates. At the semantic level, many predicates do not show obvious spatial information. For example, when "boy-watching-crafts" is known, we can not accurately infer their geometric relationship. But we can always find a similar geometric relationship for any kind of visual relationship. Thus, in order to better explain the spatial similarity of predicates, it is necessary to deal with geometric relations independently.

In summary, in order to effectively explain the similarity and difference of visual relationships, we propose a novel method - **P**rogressive **V**isual **R**elationship **I**nference(**PVRI**) - which considers both rough visual appearance and fine-grained visual cues to gradually infer visual relationships. We briefly show our ideas in fig. 1. It includes the following three steps. 1) **K**nown **C**ues **C**ollection(**KCC**): firstly, we utilize Large Language Model(LLM) to collect the cues that may help infer visual relationships; 2) **U**nknown **C**ues **E**xtraction(**UCE**): secondly, we design UCE strategy to extract the cues that are not defined by the text. 3) **P**rogressive **I**nference(**PI**): thirdly, we utilize the obtained cues to infer visual relationships.

To verify the effectiveness of our ideas, we conducted a comprehensive set of experiments on the Scene Graph Generation(SGG) task. For SGG, we extensively validate our methods on two datasets, including Visual Genome dataset(Krishna et al. (2017)), Open Image V6(Kuznetsova et al. (2020)). A series of experiments have shown that our method has achieved competitive or state-of-the-art performance on all benchmark metrics. The main contributions of our works are three-folds. 1) We analyze in detail the defects of using rough visual appearance to represent visual relationships in existing methods, and propose a novel method: **PVRI**. It considers both rough visual appearance and fine-grained visual cues to gradually infer visual relationships. 2) We utilize LLM to provide the detailed descriptions for some visual relationships. It is not only provides the accuracy of the model, but also does not limit the flexibility of the model. 3) We conduct experiments on four benchmark datasets and demonstrate the effectiveness and interpretability of our method.

## 2 RELATED WORKS

Visual relationship, which describes the interaction between subject and object, plays an important role in visual scene understanding. In recent years, there are many related researches on visual relationship(Xu et al. (2017); Li et al. (2017b); Zellers et al. (2018); Yang et al. (2018); Li et al. (2018)), which can be divided into two types: visual relationship detection and scene graph generation.

**Visual relationship detection.** Visual relationship detection(VRD) aims to detect objects in a given image and identify the interaction between them. The early works(Atzmon et al. (2016); Divvala et al. (2014); Ramanathan et al. (2015); Sadeghi & Farhadi (2011)) regard the whole subject-predicate-object triplet as a unique category for classification. Due to the long-tailed data distribution, most relationship categories suffer from the lack of sufficient training examples(Zheng et al. (2019); Li et al. (2021)). To address this problem, later works are proposed to learn the subject, object and predicate separately(Lu et al. (2016); Zhang et al. (2017); Li et al. (2017a); Yu et al. (2017)). Most of them directly extract appearance features from bounding boxes of the subject and the object of their union boxes. Although great progress has been made, the rough visual appearance is not enough to capture complex visual relationships.

**Scene graph generation.** VRD independently predicts each pair of relationships, while scene graph generation(SGG) considers that there is a correlation between all objects in the scene. In recent years, to enhance the discriminability of relationship representation, SGG attempts to design various message passing strategies(Chang et al. (2021); Li et al. (2017b); Cong et al. (2018); Zellers et al. (2018)). A popular idea is to model the context based on a sequential model(Zellers et al. (2018); Tang et al. (2019))(e.g.LSTM) or a fully-connected graph(Chang et al. (2021); Li et al. (2017b)). They utilize context information to optimize the representation of objects and predicates, and extensive research has proved the effectiveness of this idea.

**Hierarchy of visual relationships.** In recent years, many studies try to utilize a hierarchy to show the similarity and difference of predicates. Zhou et al. (2020) utilizes clustering to construct hierarchical structure of the predicates. Yang et al. (2021); Zhang et al. (2024a) distinguishes the hierarchical structure of the predicates based on their semantic meaning. Yu et al. (2020) construct the hierarchical tree structure for predicate based on the cognition. Extensive experiments have proved the effectiveness of their works. However, we recognize that it is difficult to explain the spatial similarity between predicates. At the semantic level, many predicates do not show obvious spatial information. For example, when "boy-watching-crafts" is known, we can not accurately infer their geometric relationship. But we can always find a similar geometric relationship for any kind of visual relationship. Thus in this paper, we deal with geometric relationships independently and only build the hierarchy for non-geometric predicates.

## 3 PROGRESSIVE VISUAL RELATIONSHIP INFERENCE

### 3.1 PRELIMINARIES

In this section, we first introduce the formulation of the SGG task and then briefly introduce our method. Some detailed descriptions can be found in our appendix A.1.

**Formulation.** Our main research task is scene graph generation. SGG is a task of generating a scene graph $G = \{V, E\}$ from an image, while $V$ denotes the node set consisting of objects and $E$ denotes the edge set that represents the predicates between objects. Each object entity $v_i \in V$ has an object category label $v_i^c$ from a set of object categories $C_v$ and box coordinates $v_i^b$. Each predicate $e_j \in E$ represents the $j$-th triplet $(s_j, p_j, o_j)$, where subject $s_j$ and object $o_j$ indicate related to object entities and predicate $p_j$ has a predicate category label $p_j^c$ from a set of predicate categories $C_p$. Generating $V$ and $E$ correspond to object detection and relation extraction, respectively. We give a comprehensive definition of the symbols used in this article in our appendix A.1, please consult yourself.

**Method Overview.** In this paper, in order to effectively explain the similarity and difference of visual relationships, we propose a novel method - **P**rogressive **V**isual **R**elationship **I**nference(**PVRI**) - which considers both rough visual appearance and fine-grained visual cues to gradually infer visual relationships. It includes the following three steps. 1) **K**nown **C**ues **C**ollection(**KCC**): firstly, we

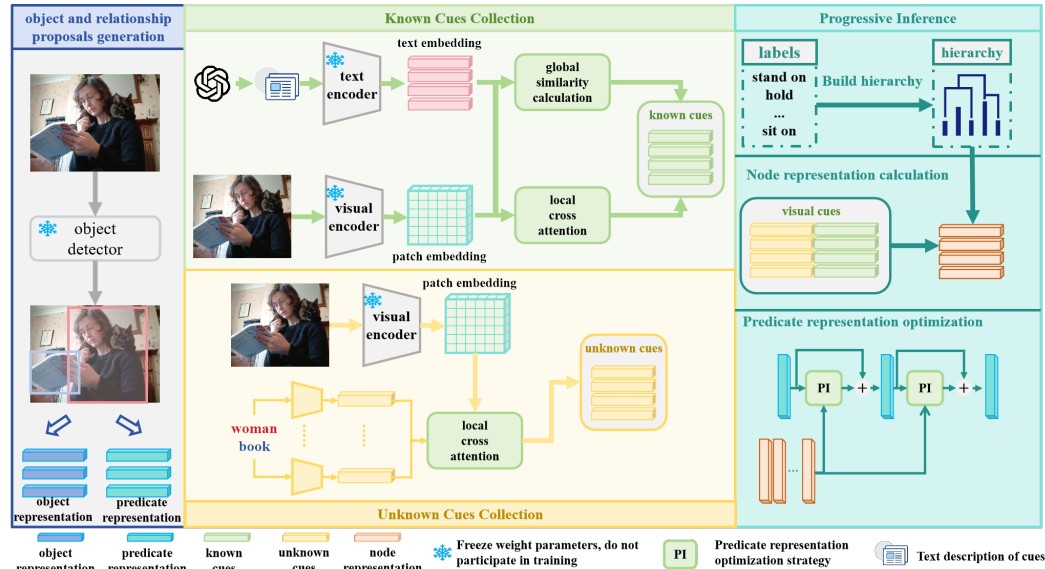

Figure 2: Illustration of overall pipeline of our PVRI. **Object and relationship proposals generation** follow the existing works and undertakes the work of the object detection; **Known cues collection** aims to extract visual representations of known cues through LLMs and VLMs; **Unknown cues extraction** aims to find more unknown cues; **Progressive Inference** aims to optimize the predicate representation through the visual cues; **LLM** used in our model is GPT-4, and **VLM** is VIT-B/16 of CLIP.

utilize Large Language Model(LLM) to collect the cues that may help infer visual relationships; 2) **U**nknown **C**ues **E**xtraction(**UCE**): secondly, we design UCE strategy to extract the cues that are not defined by the text. 3) **P**rogressive **I**nference(**PI**): thirdly, we utilize the obtained cues to infer visual relationships. In this section, we introduce our method.

## 3.2 OBJECT AND RELATIONSHIP PROPOSALS GENERATION

Our method follows the basic settings of the existing two-stage scene graph generation method. That means we first detect the objects in the image, and then predict the relationship between each pair of objects. Following (Zellers et al. (2018); Xu et al. (2017)), we first utilize an object detector(such as: Faster-RCNNRen et al. (2015)) to generate the object and relationship proposals. The object proposals are taken directly from the detection output with their categories and classification scores, while the relationship proposals are generated by forming ordered pairs of all the object proposals.

Given the relationship proposals, we calculate the representations of objects and predicates. Specifically, for the $i$-th object proposal, we denote its convolution feature as $v_i$, its bounding box as $b_i$, and its detected class as $c_i$. And then the object representation $o_i$ use a fully-connected network $f_o$ to integrate its visual, geometric and semantic features as,

$$o_i = f_o(v_i \oplus g_i \oplus emb_i) \tag{1}$$

where $\oplus$ is the concatenation operation, $g_i$ is a geometric feature based on $b_i$ and $emb_i$ is a semantic feature based on a word embedding of $c_i$.

And for the representation of predicates, considering the lack of depth information in the 2D image, we utilize the visual appearance and spatial position of the objects to predict the geometric relationship. For the relationship proposals from $i$ to $j$, we denote the convolution feature of their union-box as $u_{ij}$. Formally, we compute the initial predicate representation $rg_{ij}$ as,

$$rg_{ij} = f_{rg}(u_{ij} \oplus g_{ij}) \tag{2}$$

where $f_{rg}$ is a fully-connected network with two layers to unify the dimension and $g_{ij}$ is the spatial features which can be computed by the method in (Li et al. (2021); Zellers et al. (2018)). In our

model, this initial representation of predicate is used to predict the geometric relationship at the bottom-level of our hierarchy. And then, we will utilize the fine-grained visual cues to gradually optimize it.

## 3.3 KNOWN CUES COLLECTION

A major problem is **how to extract the fine-grained visual cues that can help infer the visual relationships**? A simple idea is to manually provide some fine-grained annotations. Due to the large numbers of visual relationship categories, it is obviously unwise to manually annotate the cues for each image. It is not only has a huge workload, but also limits the flexibility of the model. To address this problem, we propose **K**nown **C**ues **C**ollection(**KCC**) strategy to utilize the Large Language Model(LLM) to collect relevant cues. For us, these cues are already described in text, so we call them as **known cues**.

Inspired by Li et al. (2024b), we utilize LLMs to generate the textual description of the visual cues. Specifically, we design the prompts about subject and object to generate the accurate and rich descriptions of visual cues. The prompts we used can be found in appendix A.3. And then, we group them according to the category of objects. For the $i$-th object category, we denote the set of visual cues related to its categories as $TC_i \in \{TC_i^1, ..., TC_i^{m_i}\}$, where $m_i$ represents that there are $m_i$ cues related to object $i$.

Extensive research has shown that LLM contains important world knowledge. Thus, it can provide us with excellent guidance. However, we realize that these textual descriptions can not be directly used to infer visual relationships. In a visual scene, an object may interact with multiple other objects. These text descriptions are not fully applicable to the current object pair. Meanwhile, due to the factors such as occlusion, image contains the changes that are difficult to summarize in text. Thus, we need to combine the image content to filter the text.

A natural question that arises is: **how to effectively align text embedding and visual representation**? In this paper, we utilize large Vision Language Model(VLM) to address this problem. VLMs like CLIP(Radford et al. (2021)) are pretrained on web-scale datasets consisting of image-text pairs, resulting in a high degree of alignment between visual text. Many previous works have also proved its ability to capture fine-grained visual cues.

Specially, for the relationship proposal from object $i$ to $j$, we extract their clip patch embeddings $pe_{ij} \in R^{14 \times 14 \times 512}$ in their union box through the visual encoder of CLIP. And then, we input the textual descriptions of their visual cues $TC_i$ and $TC_j$ into the text encoder to obtain the corresponding text embeddings $te_{ij} \in \{te_{ij}^1, ..., te_{ij}^{(m_i+m_j)}\}$. Followed by, we treat text embeddings as a set of queries, and perform global similarity calculation and local cross-attention with patch embeddings, respectively.

**Global similarity calculation.** Given the relationship proposals from object $i$ to $j$, we utilize a convolutional network $f_g$ to extract their global representation $gv_{ij}$,

$$gv_{ij} = f_g(pe_{ij}) \tag{3}$$

and then we calculate the similarity $sim_{ij} \in \{sim_{ij}^1, ..., sim_{ij}^{(m_i+m_j)}\}$ between each text embedding and it. For the $k$-th text embedding $te_{ij}^k$, its similarity to $gv_{ij}$ is defined as $sim_i^k j$,

$$sim_{ij}^k = cos\_sim(gv_{ij}, te_{ij}^k) \tag{4}$$

where $cos\_sim$ is the cosine similarity method. Each similarity value represents the degree of correlation between the visual cue and the current object pair.

**Local cross attention.** Every text embedding are treated as a query to perform a convolution operation on patch embeddings $pe_{ij}$ and get the spatial activation map $sp_{ij} \in \{sp_{ij}^1, ..., sp_{ij}^{(m_i+m_j)}\}$,

$$sp_{ij}^k = sigmoid(pe_{ij} \odot te_{ij}^k) \tag{5}$$

where $\odot$ represents the convolution operation. Each value in the spatial active map represents how likely this local region contains the corresponding visual cues.

After the above convolutional operation, for each patch embedding we get s set of spatial activation maps corresponding to text embeddings. Then we utilize these spatial activation maps to perform

region-based attention and weighted average pooling on original patch embeddings. We utilize the activation values as the the pooling weights. Therefore, we can get $N$ visual embeddings corresponding to $N$ text embeddings. For the $k$-th visual embedding $vc_{ij}^k$, it can be computed by,

$$vc_{ij}^k = gap(pe_{ij} \otimes sp_{ij}^k) \tag{6}$$

where $\otimes$ denotes elements-wise product, $gap$ is the global average pooling. Finally, their known cues $kc_{ij}$ can be computed by,

$$kc_{ij} = sim_{ij} \cdot vc_{ij} \tag{7}$$

Through the above operations, we first utilize LLMs to collect known cues. And then we measure their existence in the image through **global similarity calculation** strategy, and capture the visual representation of these cues through **local cross attention** strategy. However, we realize that these cues may not be complete.

### 3.4 Unknown Cues Extraction

Visual perception is a rich signal for modeling a vastness of experiences in the world that cannot be documented by text alone. In fact, it is difficult for us to fully summarize the cues on which a visual relationship depends. For example, although the "upright leg" is the key factor of our inference of "stand on", the "upright body" may also be used as a basis for our judgment when the leg is occluded. Thus, text description can be used as an excellent guidance, but should not be used as the only criterion. We call these cues that are not/are difficult to describe in text as **unknown cues**. Our model must have the ability to extract unknown cues from images.

Since the visual relationship depends on the object, we believe that the information that helps to infer the visual relationship should be at least related to the object. Therefore, we perform a simple decoupling of the object. As shown in fig.2, we generates $P$ convolutional filters independently based on class semantic knowledge. Concretely, for the $i$-th object proposal, we get its class semantic knowledge $ck_i$ of its category through clip text encoder. And then, we design $P$ different MLPs to decouple it. Every MLP independently maps the class semantic vector from semantic space to a $1 \times 1$ convolutional filter in visual space.

Followed by, for the relationship proposal from object $i$ to $j$, we can get $2 \times P$ latent parts. We treat them as queries to perform **local cross attention** similar to KCC to get the set of unknown cues $uc_{ij} \in \{uc_{ij}^1, ..., uc_{ij}^{2 \times P}\}$. It is worth mentioning that because it is based on the decoupling of the object, it is not necessary to judge whether it exists in the image. In other words, there is no **global similarity calculation**. We will utilize them in the next step. The details of our UCE strategy can be found in our appendix A.4.

Through the above operations, we captured the visual cues that may help to infer the visual relationship of the current object pair through **KCC** and **UCE** strategies. For the relationship proposal from object $i$ to $j$, the visual cues we get include two parts: $kc_{ij}$(**known cues**) and $uc_{ij}$(**unknown cues**). And then, we will utilize these visual cues to progressively optimize predicate representation.

### 3.5 Progressive Inference

In general, our progressive inference strategy is based on geometric relationships and gradually optimizes predicate representation. It consists of two parts: for geometric relationship, its predicate representation follows the existing works, that is the initial predicate representation we mentioned in section 3.2; for non-geometric relationship, its predicate representation can be constructed by the following steps.

**Build hierarchy.** Firstly, we construct a hierarchy to reflect the similarity of predicates. Following the existing works(Zhang et al. (2024a); Wang et al. (2019)), we construct this hierarchy according to the semantic similarity of predicates. Specifically, according to the semantic embedding of predicates, we utilize **hierarchical clustering** strategy to build this hierarchy. The detailed description can be found in appendix A.2.

**Node representation calculation.** In this hierarchy, the nodes in the last layer are meaningful predicate labels. We can easily get their representations. However, for other nodes in the hierarchy, they have no practical significance and are only generated during our clustering process. Thus, we

first calculate the representation of these nodes. Suppose that the $k$-th node of the $l$-th layer has $n$ child nodes in the $l+1$-th layer. And its node representation $hn_l^k$ is the average value of its child node representations,

$$hn_l^k = \frac{1}{n} \sum_{j=1}^{n} hn_{l+1}^j \tag{8}$$

If the node has no subsequent nodes, it indicates that the node is a specific predicate, and the representation of this node can be calculated by the following steps. Firstly, we get the text embedding of all predicates $tp$ by clip text encoder. And then for the relationship proposal from object $i$ to $j$, we calculate the similarity $ps_{ij}$ between each predicate and the visual cues obtained by our **KCC** and **UCE** strategies,

$$ps_{ij}^k = cos\_sim((kc_{ij} \oplus uc_{ij}), tp^k) \tag{9}$$

Finally, the node representation for the last layer in our hierarchy $hn_{la}$ can be computed by,

$$hn_{la} = tp + ps_{ij} \cdot (kc_{ij} \oplus uc_{ij}) \tag{10}$$

It is worth mentioning that in the clustering process of our hierarchy, we utilize the word embedding from **glove**, while the computational node representation utilizes the text embedding from **clip**. The specific reasons can be found in our appendix A.2.

**Predicate representation optimization.** For the non-geometric predicates, their initial representations are the visual embedding from the clip visual encoder. We will optimize them according our hierarchy. Given the relationship proposal from object $i$ to $j$, we denote their predicate representation in $l$-th layer as $re_{ij}$. Followed by, we can calculate the similarity between it and all nodes in $l+1$-th layer as $sh_{ij}^{l+1}$,

$$sh_{ij}^{l+1} = softmax(\frac{re_{ij}^l \cdot hn_{l+1}}{d_k}) \tag{11}$$

where $d_k$ is the dimension of these embeddings(in this work, its value is 512). Then, the predicate representation in $l+1$ can be computed by,

$$re_{ij}^{l+1} = re_{ij}^l + sh_{ij}^{l+1} \cdot hn_{l+1} \tag{12}$$

Finally, for the relationship proposal from object $i$ to $j$, their final predicate representation $r_{ij}$ can be computed by,

$$r_{ij} = f_r(rg_{ij} \oplus re_{ij}^{la}) \tag{13}$$

where $f_r$ is a fully connected network to unify the dimension and $re_{ij}^{la}$ is the last representation through the above optimization process.

### 3.6 MESSAGE PASSING, PREDICTOR AND TRAINING LOSS.

**Message Passing.** Message Passing(MP) aims to build connections between entities at the same level, which optimizes object and predicate representation through the context information of the scene. There have been extensive studies to prove the effectiveness of this strategy. In this work, our message passing strategy follows the setting of BGNNLi et al. (2021). Concretely, for each object and predicate representation, the final representation of them is denoted as $o^+$ and $r^+$.

**Predictor.** To predict the object and predicate, we introduce two linear classifiers. For predicate, our classifier integrates the final representation of predicate $r_{ij}^+$ and a class frequency $q_{ij}$ prior for classificationZellers et al. (2018). The distribution of predicate is denoted as $p_{r_{ij}}$,

$$p_{r_{ij}} = softmax(W_{rel}r_{ij}^+ + q_{ij}) \tag{14}$$

where $W_{rel}$ is the parameter of predicate classifier.

For object, our classifier takes as input the final object representation $o_{ij}^+$. The distribution of object is denoted as $p_{o_{ij}}$,

$$p_{o_i} = softmax(W_{obj}o_i^+) \tag{15}$$

where $W_{obj}$ is the parameter of object classifier.

**Training Loss.** To train our PVRI model, we design a multi-tasks loss of three components, including $L_p$ for predicate classification, $L_o$ for object classification, $L_{pi}$ for **PI** strategy. Formally,

$$\mathcal{L}_{total} = \mathcal{L}_p + \lambda_o \mathcal{L}_o + \lambda_{pi} \mathcal{L}_{pi} \tag{16}$$

where $\lambda_o$, $\lambda_{rti}$ are weight parameters for calibrating the supervision from each sub-task, $\mathcal{L}_p$, $\mathcal{L}_o$, are the standard cross entropy loss for multi-class classification(foreground categories plus background) and $\mathcal{L}_{pi}$ is cosine similarity loss, which is used to measure the similarity between the predicate representation of each layer and the representation of all nodes in the layer in our **PI** strategy.

## 4 EXPERIMENTS

### 4.1 EXPERIMENTS CONFIGURATIONS

**Dataset Details.** To verify the effectiveness of our method, we conduct experiments on a variety of datasets, including Visual Genome dataset(Krishna et al. (2017)), Open Image V6(Kuznetsova et al. (2020)).

*Visual Genome(VG).* **VG** is the most commonly used dataset in SGG task. It consists of 108,073 images, including tens of thousands of unique object and predicate categories. However, most categories have a very limited number of instances. In our experiments, we follow the most commonly used data splits proposed by (Xu et al. (2017); Zellers et al. (2018)). The 150 most frequent object categories and the 50 most frequent predicate types are adopted for evaluation.

*OpenImages V6(OI).* **OI** dataset is a large scale dataset commonly used for SGG tasks. It contains a diverse collection of over 133k images with 126368 training, 1813 validation, and 5322 testing images. This dataset covers a wide range of real-world scenarios. The OI provides object-level annotations for each image, including bounding-boxes and 301 object categories. In addition, it includes 31 relationship annotations that describe the interactions and connections between pairs of objects within a scene.

**Evaluation Protocol.** For VG dataset, we evaluate the model on three sub-tasks as Xu et al. (2017); Zellers et al. (2018): 1) **predicate classification (PredCls)**: Given the ground-truth bounding boxes and class labels of objects, we need to predict the visual relationship classes among the object pairs. 2) **scene graph classification (SGCls)**: Given the ground-truth bounding boxes of objects, we need to predict both the object and predicate classes. 3) **scene graph generation (SGGen, also denote as SGDet)**: Given an image, we need to detect the objects and predict their pairwise relationship classes. In each task, following the previous works Zellers et al. (2018); Li et al. (2021); Lin et al. (2020), we takes **recall@K(R@K)** and mean **recall@K(mR@K)** as evaluation metrics.

And for OpenImages V6 dataset, in addition to **R@K** and **mR@K**, we employed the following three additional metrics to provide a more comprehensive assessment of SGG methods: 1) **Weighted Mean Average Precision for Relationships**($wmAP_{rel}$) evaluates the performance of the model in predicting the relationships between object pairs. It calculates the mean AP for each relationship category, weighted by the number of ground-truth instances of that relationship in a dataset. It provides a more balanced evaluation by considering the varying frequencies of different relationship types in scene graphs. 2) **Weighted Mean Average Precision for Phrases** ($wmAP_{phr}$) assess the ability of the model to predict relationship phrases involving both object categories and their corresponding relationships. 3) **Weighted Score**($score_{wtd}$) is a comprehensive evaluation metric that combines the performance of the model with both the relationship and phrase predictions, considering their relative importance in scene graphs. This is the weighted sum of $wmAP_{rel}$ and $wmAP_{phr}$, where the weights are determined based on the significance of the relationships and phrases in a dataset. $score_{wtd}$ was calculated as: $score_{wtd} = 0.2R@50 + 0.4wmAP_{rel} + 0.4wmAP_{phr}$.

**Implementation details.** Our model is similar to the existing two-stage model. In the first stage, we utilize the Faster-RCNN as the object detector, which is based on ResNeXt-101-FPN backbone provided by Xie et al. (2017). And in the second stage, we utilize VIT-B/16 of CLIP model as the backbone to extract the visual cues. The size of patch embedding is 14×14×512. All of our experiments were performed on three 3090 GPUs. The batch size and initial learning rate are set to 9 and 0.024, respectively. Our model is optimized by the Adam algorithm with the momentums of 0.9 and 0.999. The division of datasets follows the most common strategy in the field.

Table 1: The performance of state-of-the-art SGG models on three SGG tasks with graph constraints setting on mR@50/100 and R@50/100 on the VG dataset. The **best** method is marked according to formats.

| Models | PredCls | | SGCls | | SGGen | |
|---|---|---|---|---|---|---|
| | mR@50/100 | R@50/100 | mR@50/100 | R@50/100 | mR@50/100 | R@50/100 |
| IMP(Xu et al. (2017)) | 9.8/10.5 | 59.3/61.3 | 5.8/6.0 | 34.6/35.4 | 3.8/4.8 | 20.7/24.5 |
| MOTIFS(Zellers et al. (2018)) | 14.0/15.3 | 65.2/67.1 | 7.7/8.2 | 35.8/36.5 | 5.7/6.6 | 27.2/30.3 |
| VCTree(Tang et al. (2019)) | 17.9/19.4 | 66.4/68.1 | 10.1/10.8 | 38.1/38.8 | 5.9/8.0 | 27.9/31.3 |
| Unbiased(Tang et al. (2020)) | 25.4/28.7 | 47.2/51.6 | 12.2/14.0 | 25.4/27.9 | 9.3/11.1 | 19.4/23.2 |
| MSDN(Li et al. (2017b)) | 19.2/20.5 | 65.0/66.7 | 11.6/12.6 | 38.9/39.8 | 7.7/9.0 | 30.3/33.3 |
| GPS-Net(Lin et al. (2020)) | 15.2/16.6 | 65.2/67.1 | 8.5/9.1 | 37.8/39.2 | 6.7/8.6 | **31.1/35.9** |
| GB-NET | 22.1/24.0 | 66.6/68.2 | 12.7/13.4 | 37.3/38.0 | 7.1/8.5 | 26.3/29.9 |
| SMN(Zellers et al. (2018)) | 13.3/14.8 | 65.2/67.1 | 7.1/7.6 | 35.8/36.5 | 5.3/6.1 | 27.2/30.3 |
| BGNN(Li et al. (2021)) | 30.4/32.9 | 59.2/61.3 | 14.3/16.5 | 37.4/38.5 | 10.7/12.6 | 31.0/_35.8_ |
| PENET(Zheng et al. (2023)) | 31.5/33.8 | **68.2/70.1** | 17.8/18.9 | **39.4/40.7** | 12.4/14.5 | 30.7/35.2 |
| HetSGG(Yoon et al. (2023)) | 31.6/33.5 | 57.8/58.9 | 17.2/18.7 | 37.6/38.5 | 12.2/14.4 | 30.0/34.6 |
| EdgeSGG(Kim et al. (2023)) | 34.7/36.9 | 60.1/61.8 | 17.8/18.8 | 39.1/40.1 | 13.6/15.8 | 29.7/34.0 |
| PVRI(Ours) | **36.9/38.2** | 55.4/57.1 | **18.7/20.6** | 35.2/36.1 | **15.4/17.2** | 28.2/30.9 |

Table 2: The SGG performance of SGGen with graph constraints for the Head, Body and Tail. The **best** and _second best_ methods are marked according to formats.

| Models | PredCls | | SGCls | | SGGen | | R@100 | | |
|---|---|---|---|---|---|---|---|---|---|
| | mR@50/100 | R@50/100 | mR@50/100 | R@50/100 | mR@50/100 | R@50/100 | Head | Body | Tail |
| GPS-NetLin et al. (2020)(2020) | 15.2/16.6 | **65.2/67.1** | 8.5/9.1 | **37.8/39.2** | 6.7/8.6 | **31.1/35.9** | _30.8_ | 8.5 | 3.9 |
| VCTreeTang et al. (2019)(2019) | 17.9/19.4 | **66.4/68.1** | 10.1/10.8 | 38.1/38.8 | 5.9/8.0 | 27.9/31.3 | 24.7 | 12.2 | 1.8 |
| BGNNLi et al. (2021) | **30.4/32.9** | 59.2/61.3 | _14.3/16.5_ | 37.4/38.5 | 10.7/12.6 | _31.0/35.8_ | **34.0** | _12.9_ | _6.0_ |
| PVRI(Ours) | 36.9/38.2 | 55.4/57.1 | 18.7/20.6 | 35.2/36.1 | **15.4/17.2** | 28.2/30.9 | 27.7 | **21.1** | **10.2** |

## 4.2 QUANTITATIVE EXPERIMENTS

**Baselines.** In this section, we compare our proposed method with several existing state-of-the-art methods: IMPXu et al. (2017), MOTIFSZellers et al. (2018), UnbiasedTang et al. (2020), VC-TreeTang et al. (2019), SMNZellers et al. (2018), GB-Net, MSDNLi et al. (2017b), BGNNLi et al. (2021), GPS-NetLin et al. (2020), PPDLLi et al. (2022b), Nice-MotifLi et al. (2022a), PENetZheng et al. (2023), HetSGGYoon et al. (2023) and EdgeSGGKim et al. (2023).

**Comparison with overall performance.** The quantitative results are reported in table 1. Our method shows superior performance on all subtasks on metrics, especially on **mR@K**. Because the VG dataset has an imbalanced data distribution, mR@K, which prefers tail predicates, can be said to be more reliable than R@K metrics that focus on common predictions with abundant samples. For **PredCls**, PVRI achieves **36.9** on mR@50 and **38.2** on mR@100, indicating its effectiveness and generic capture of more relevant predicates within the top-50 and top-100 predications, respectively. Similarly, for **SGCls**, our PVRI achieves **18.7** on mR@50 and **20.6** on mR@100, respectively. More importantly, for **SGGen** with more noise interference, our PVRI also shows excellent performance. Our method achieves **15.4** on mR@50 and **17.2** on mR@100. Meanwhile, the performance on OpenImages V6 are reported in table 3. These results prove that our method can effectively reduce the ambiguity of visual relationship representation.

**The results on different predicate groups.** In addition, we also report the performance of our model on different predicate groups. Following the similar protocol in Liu et al. (2019), we divide the categories into three disjoint groups according to the instance number in training split: _head_ (more than 10k), _body_ (0.5k ∼ 10k), _tail_ (less than 0.5k). The results are shown in table 2. Our method achieves **21.1%** and **10.2%** on body and tail, respectively. It indicates that even for less common predicates, our method still has excellent recognition ability. These results demonstrate that **our method is robust to visual relationships**.

Table 3: Performance comparison with the SoTA methods on OpenImage V6 dataset. The **best** method is marked according to formats

| Method | mR@50 | R@50 | $wmAP_{rel}$ | $wmAP_{phr}$ | $score_{wtd}$ |
|---|---|---|---|---|---|
| VCTree(2019) | 33.9 | 74.1 | 34.2 | 33.1 | 40.2 |
| RelDN(2019) | 37.2 | 75.3 | 32.2 | 33.4 | 42.0 |
| BGNN(2021) | 40.5 | 75.0 | 33.5 | 34.1 | 42.1 |
| HetSGG(2023) | 42.7 | 76.8 | 34.6 | 35.5 | 43.3 |
| PEBET(2023) | - | 76.5 | 36.6 | 37.4 | 44.9 |
| EdgeSGG(2023) | 43.3 | **77.1** | 36.4 | 37.4 | 44.9 |
| Ours | **43.9** | 76.0 | **36.9** | **37.6** | **45.2** |

### 4.3 ABLATION STUDY

The core of our method is the process of progressive inference combined with visual cues. In the whole process, we utilize two strategies to capture visual cues: **KCC** and **UCE**. **KCC** aims to capture visual cues through LLMs, while **UCE** aims to enable the model to discover the visual cues by itself. In this section, we verify their effectiveness, respectively.

Table 4: Performance of ablation study. The **best** method is marked according to formats.

| Models | SGGen | | | | |
|---|---|---|---|---|---|
| | mR@100 | R@100 | Head | Body | Tail |
| VCTree | 8.0 | 31.3 | 24.7 | 12.2 | 1.8 |
| GPS-Net | 8.6 | 35.9 | 30.8 | 8.5 | 3.9 |
| BGNN | 12.6 | **35.8** | **34.0** | 12.9 | 6.0 |
| PVRI(Ours) | **17.2** | 30.9 | 27.7 | **21.1** | **10.2** |
| $KCC_{only}$(Ours) | 16.1 | 29.1 | 26.2 | 20.7 | 9.7 |
| $UCE_{only}$(Ours) | 14.4 | 31.1 | 28.3 | 16.2 | 8.6 |

**Experiments with only KCC retained.** We first evaluate the importance of our **KCC** strategy. The text description generated by LLM can provide excellent guidance for us to understand the visual relationship. As shown in table 4, for SGGen task, this strategy achieves **16.1** on **mR@100** and **29.1** on **R@100**. For different predicate groups, this strategy achieves **20.7** on Body and **9.7** on Tail. Compared with the complete model, this strategy has declined in all aspects of indicators. This is because some predicates(e.g., "of") do not depend on some certain specific cues.

**Experiments with only UCE retained.** And then we evaluate the importance of our **UCE** strategy. **UCE** aims to enable the model to discover the visual cues by itself. As shown in table 4, for SGGen task, this strategy achieves **14.4** on **mR@100** and **31.1** on **R@100**. For different predicate groups, this strategy achieves **16.2** on Body and **8.6** on Tail. It can be seen that this strategy is effective, but it is not the main reason for our performance improvement.

## 5 CONCLUSION

In general, in order to solve the shortcomings of the existing method to represent predicates, we propose a novel method - **P**rogressive **V**isual **R**elationship **I**nference(**PVRI**) - which considers both rough visual appearance and fine-grained visual cues to gradually infer visual relationships. In order to prove the effectiveness of our method, we conducted a large number of experiments on Visual Genome, OpenImages datasets. Various experimental results prove the effectiveness and versatility of our method. As an important part of scene understanding, the accuracy and interpretability of visual relationship are very important for us to understand the scene. We also look forward to more research attempts to better collect visual cues.

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

# A  APPENDIX

In order to better illustrate the main content of our method, our appendix contains the following seven aspects:

1) Problem Setting and Related Works(appendix A.1). In this subsection, we introduce our **problem settings**, **related works**, **method overview**, **dataset details** and **symbol setting**;

2) Hierarchy of Visual Relationships(appendix A.2). In this subsection, we introduce our hierarchy in detail from the **existing works**, **existing problems**, **our methods**, **selection of node representations**, **results on different datasets**, and **comparison with existing methods**;

3) Known Cues Collection(appendix A.3). In this subsection, we introduce our KCE strategy in detail from the **prompts**, **visualization of known cues**, and **specific steps**;

4) Unknown Cues Extraction(appendix A.4). In this subsection, we introduce our UCE strategy in detail from the **specific steps**;

5) Progressive Inference(appendix A.5). In this subsection, we introduce our PI strategy in detail from the **specific steps** and **some examples**.

7) Summary and Limitations(appendix A.6). In this subsection, we summarize our ideas and propose possible modification directions.

## A.1  PROBLEM SETTING AND RELATED WORKS

**Problem setting.** We aim to tackle scene graph generation(SGG), which parses an input image into a structural graph representation of objects and their visual relationship in the scene. SGG aims to identify objects and their predicates in a given image. Given an image I, the purpose of SGG is to parse I into a scene graph G,

$$G = V_o, E_r \tag{17}$$

where $V_o$ is the node set of object and $E_r$ is the edge set representing the predicate between ordered object pairs. In general, each node has a category label from a set of object classes, while each edge has a predicate class.

**Related works.** Visual relationship, which describes the interaction between subject and object, plays an important role in visual scene understanding. In recent years, there are many related researches on visual relationship(Xu et al. (2017); Li et al. (2017b); Zellers et al. (2018); Yang et al. (2018); Li et al. (2018)), which can be divided into two types: visual relationship detection and scene graph generation.

**Visual relationship detection.** Visual relationship detection(VRD) aims to detect objects in a given image and identify the interaction between them. The early works(Atzmon et al. (2016); Divvala et al. (2014); Ramanathan et al. (2015); Sadeghi & Farhadi (2011)) regard the whole subject-predicate-object triplet as a unique category for classification. Due to the long-tailed data distribution, most relationship categories suffer from the lack of sufficient training examples(Zheng et al. (2019); Li et al. (2021)). To address this problem, later works are proposed to learn the subject, object and predicate separately(Lu et al. (2016); Zhang et al. (2017); Li et al. (2017a); Yu et al. (2017)). Most of them directly extract appearance features from bounding boxes of the subject and the object of their union boxes. Although great progress has been made, the rough visual appearance is not enough to capture complex visual relationships.

**Scene graph generation.** VRD independently predicts each pair of relationships, while scene graph generation(SGG) considers that there is a correlation between all objects in the scene. In recent years, to enhance the discriminability of relationship representation, SGG attempts to design various message passing strategies(Chang et al. (2021); Li et al. (2017b); Cong et al. (2018); Zellers et al. (2018)). A popular idea is to model the context based on a sequential model(Zellers et al. (2018); Tang et al. (2019))(e.g.LSTM) or a fully-connected graph(Chang et al. (2021); Li et al. (2017b)). They utilize context information to optimize the representation of objects and predicates, and extensive research has proved the effectiveness of this idea.

Table 5: Objects and predicates in the VG.

| Visual Genome | | |
|---|---|---|
| **Objects** | | |
| Category | Examples | Classes |
| Artifact | arm, tail, wheel | 32 |
| Person | boy, kid, woman | 13 |
| Clothes | cap, jean, sneaker | 16 |
| Vehicle | airplane, bike, truck | 12 |
| Flora | flower, plant, tree | 3 |
| Location | beach, room, sidewalk | 11 |
| Furniture | bed, desk, table | 9 |
| Building | building, house | 2 |
| Structure | fence, post, sign | 3 |
| Food | banana, orange, pizza | 6 |
| Part | arm, tail, wheel | 32 |
| **Predicates** | | |
| Geometric | above, behind, under | 12 |
| Possessive | has, part of, wearing | 8 |
| Semantic | carrying, eating, using | 24 |
| Misc | for, from, made of | 3 |

**Hierarchy of visual relationships.** Our method is implemented under the guidance of a built hierarchy. In recent years, there have many studies on the hierarchy of visual relationships(Zhou et al. (2020); Zhang et al. (2024a)). Zhou et al. (2020) utilizes clustering to construct hierarchical structure of the predicates. Yang et al. (2021); Zhang et al. (2024a) distinguishes the hierarchical structure of the predicates based on their semantic meaning. Yu et al. (2020) construct the hierarchical tree structure for predicate based on the cognition.

Unlike them, our PVRI strategy is based on the geometric relationships, considering the rough visual appearance and the detailed visual cues, and explains the similarities and differences of predicates in the hierarchical structure.

**Method Overview.** Guided by our hierarchy, we propose a novel method - **P**rogressive **V**isual **R**elationship **I**nference (**PVRI**) - which utilize the visual cues to explain the similarities and differences of visual relationships at different levels. An overview of our method shown in **fig.3**. It includes the following three steps. 1) Object and relationship proposals generation(section 3.2): we first generate object and relationship proposals; 2) **K**nown **C**ues **C**ollection(**KCC**)(section 3.3): and then, we utilize Large Language Model(LLM) to collect the cues that may help infer visual relationships; 3) **U**nknown **C**ues **E**xtraction(**UCE**)(section 3.4): followed by, we design UCE strategy to extract the cues that are not defined by the text. 4) **P**rogressive **I**nference(**PI**)(section 3.5): finally, we utilize the obtained cues to infer visual relationships. In summary, our PVRI strategy is based on the geometric relationships, considering the rough visual appearance and the detailed visual cues, and explains the similarities and differences of predicates in the hierarchical structure. In this section, we introduce our method.

**Dataset Details.** To verify the effectiveness of our method, we conduct experiments on a variety of datasets, including Visual Genome dataset(Krishna et al. (2017)), Open Image V6(Kuznetsova et al. (2020)). In this section, we make a brief introduction.

**Visual Genome(VG). VG** is the most commonly used dataset in SGG task. It consists of 108,073 images, including tens of thousands of unique object and predicate categories. However, most categories have a very limited number of instances. In our experiments, we follow the most commonly used data splits proposed by (Xu et al. (2017); Zellers et al. (2018)). The 150 most frequent object categories and the 50 most frequent predicate types are adopted for evaluation.

**OpenImages V6(OI). OI** dataset is a large scale dataset commonly used for SGG tasks. It contains a diverse collection of over 133k images with 126368 training, 1813 validation, and 5322 testing images. This dataset covers a wide range of real-world scenarios. The OI provides object-level annotations for each image, including bounding-boxes and 301 object categories. In addition, it

Table 6: The symbols used in our paper and their explanations.

| symbols | meanings |
|---|---|
| $o_i$ | the object representation for the current object |
| $rg_{ij}$ | the initial representation for the current object pairs |
| $v_i$ | the convolution feature for the current object |
| $g_i$ | the geometric features for the current object |
| $emb_i$ | the semantic embedding of object category for the current object |
| $u_{ij}$ | the convolution feature of the union-box of the current object pairs |
| $TC_i$ | the text descriptions related to current object collected from LLM |
| $te_i$ | the text embedding of $TC_i$ via the text encoder of CLIP |
| $pe_{ij}$ | the patch embedding of the relationship proposal from object $i$ to $j$ via the visual encoder of CLIP |
| $gv_{ij}$ | the global representation of object pairs |
| $cos_s im$ | cosine similarity method |
| $sp_{ij}$ | the spatial activation map of visual cues |
| $kc_{ij}$ | known cues of current object pairs |
| $uc_{ij}$ | unknown cues of current object pairs |
| $hn_l$ | the node representations of the $l$-th layer in our hierarchy |
| $CM$ | the correlation matrix of visual cues and nodes |
| $re_{ij}^l$ | the representation of predicate of the $l$-th layer |

includes 31 relationship annotations that describe the interactions and connections between pairs of objects within a scene.

**Symbol Setting.** In order to facilitate understanding, we aggregate all the symbols and explain their meanings in table 6.

## A.2 HIERARCHY OF VISUAL RELATIONSHIPS

In this subsection, we will introduce our hierarchy in detail from the **existing works**, **existing problems**, **our methods**, **selection of node representations**, **results on different datasets**, and **comparison with existing methods**;

**Existing works.** Our method is implemented under the guidance of a built hierarchy. In recent years, there have many studies on the hierarchy of visual relationships(Zhou et al. (2020); Zhang et al. (2024a)). Zhou et al. (2020) utilizes clustering to construct hierarchical structure of the predicates. Yang et al. (2021); Zhang et al. (2024a) distinguishes the hierarchical structure of the predicates based on their semantic meaning. Yu et al. (2020) construct the hierarchical tree structure for predicate based on the cognition.

Most of them try to utilize the hierarchy to explain the semantic overlap between some predicates. For example, "sitting next to" and "standing next to" are two different predicate labels but express similar spatial relations in semantics. Extensive experiments demonstrate the effectiveness of their method.

**Existing problems.** It is no doubt that the semantic embedding of predicate category is effective. In previous studies, Zhou et al. (2020) summarized the **phrase-format** predicate labels into three categories: **verb-prep**(e.g. "walking by"), **prep-prep**(e.g."in between") and **stereotyped expression**(e.g. "inside of"). It can be seen that these predicates are more or less similar to the parts that constitute them. Thus, the clustering method based on semantic embedding can well reflect the similarity between predicates to a certain extent.

However, we recognize that this method is difficult to explain the spatial similarity between predicates. At the semantic level, many predicates do not show obvious spatial features. For example, when "boy-watching-crafts" is known, we can not accurately infer their geometric relationship. But we can always find a similar geometric relationship for any kind of visual relationship.

As shown in fig. 3, in a visual scene, no matter what visual relationship a pair of objects has, we can always find a similar geometric relationship for them. It can be said that geometric relationship is the most basic visual relationship. Thus, to effectively explain the spatial similarity between predicates, it is necessary to deal with geometric relationships independently.

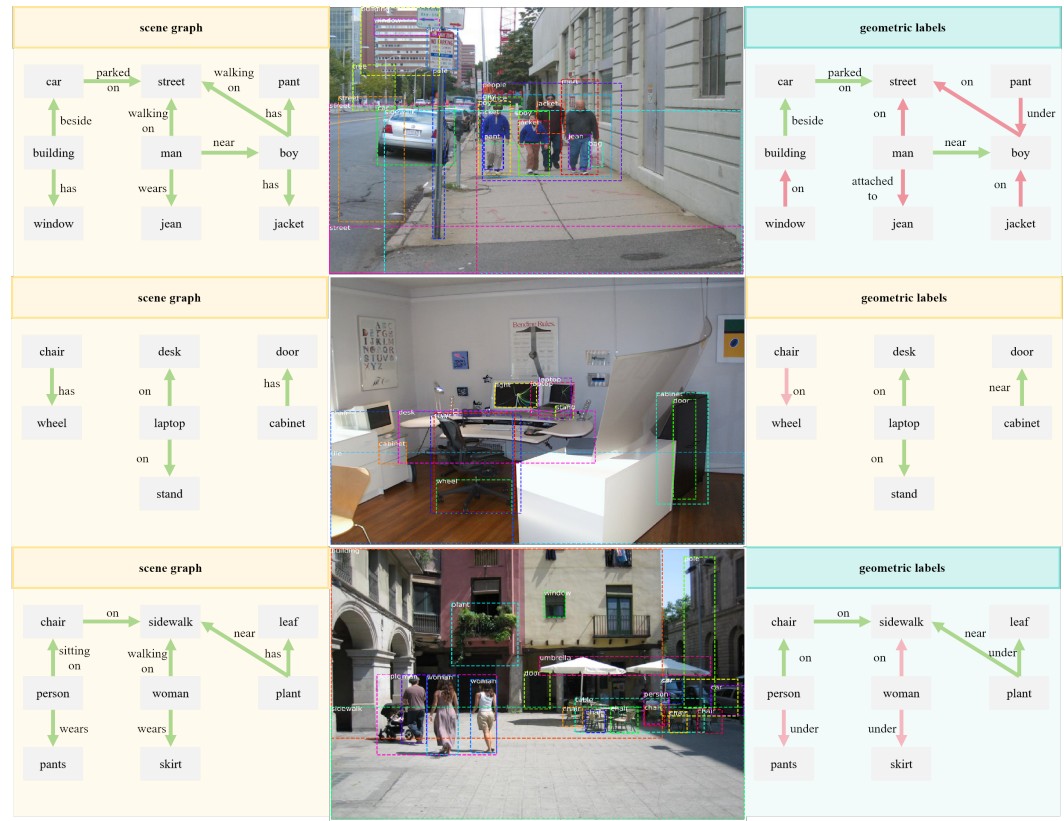

Figure 3: The scene graph of visual scene and corresponding geometric labels.

**Our method.** Specifically, our hierachy is built through the following steps. For non-geometric predicates, we utilize a hierarchical clustering algorithm to find their hierarchical structure according to their semantic similarity. This method clusters the predicate labels based on the machine understanding through utilizing the pretrained word2vec model. Each predicate label is encoded as one 300-dimensional embedding vector with word2vec pretrained model. Then, they are clustered based on their cosine similarity. We show the correlation matrix of predicates in **VG** dataset in fig. 4.

**Selection of node representations.** As we said above, we utilize the semantic embedding from glove to cluster predicates. However, the construction of node representation selects text embedding from CLIP. The main reason about that is our visual cues are captured through CLIP. The highly aligned visual and text can help us better match visual cues related to nodes. In the clustering process, we utilize semantic embedding from glove. The main reason is the difference between vision and language. The text embedding from CLIP is highly aligned with visual representation. But the visual representation of the predicate mainly depends on the visual appearance of the subject and the object. Thus, the robustness of semantic embedding makes it a better choice.

## A.3 Known Cues Collection

A major problem is **how to extract the fine-grained visual cues that can help infer the visual relationships**? A simple idea is to manually provide some fine-grained annotations. Due to the large numbers of visual relationship categories, it is obviously unwise to manually annotate the cues for each image. It is not only has a huge workload, but also limits the flexibility of the model. To address this problem, we propose **K**nown **C**ues **C**ollection(**KCC**) strategy to utilize the Large Language Model(LLM) to collect relevant cues. For us, these cues are already described in text, so we call them as **known cues**.

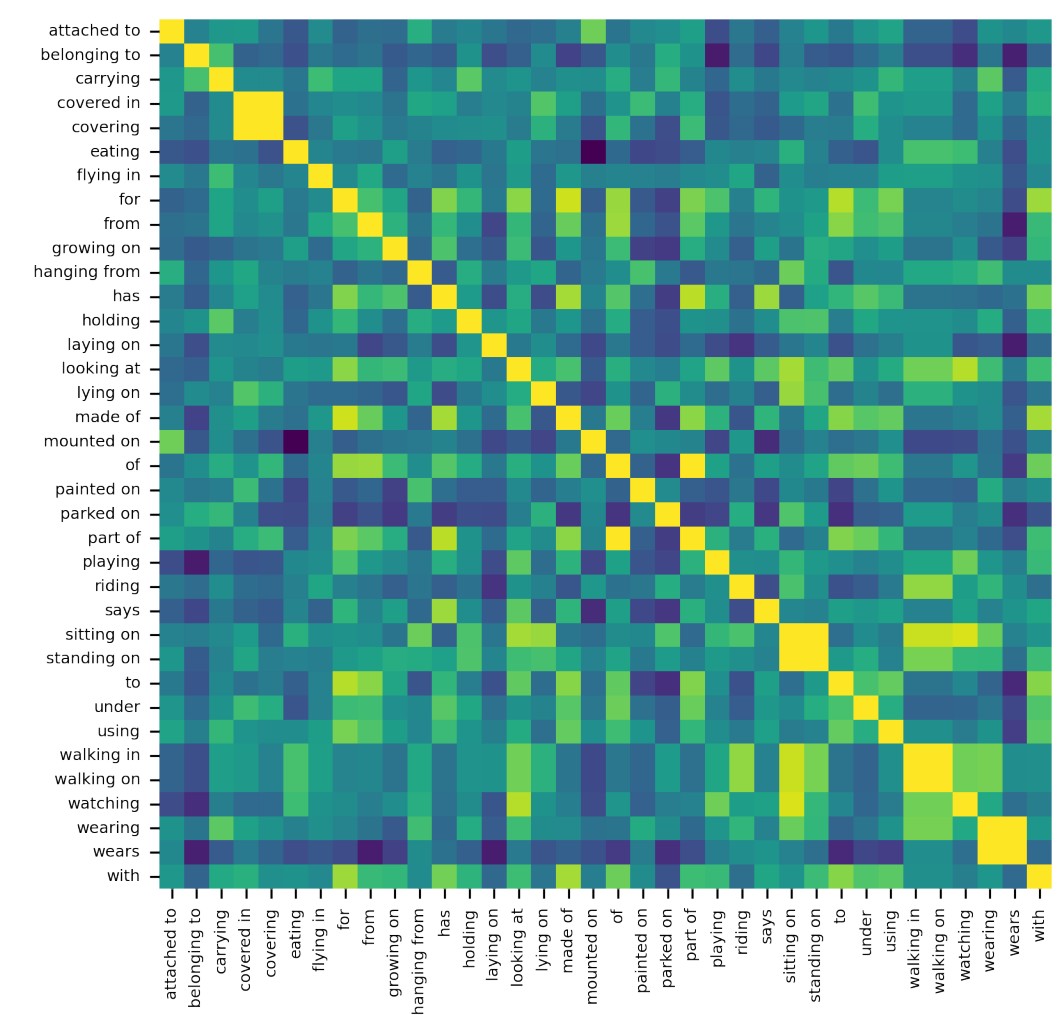

Figure 4: The scene graph of visual scene and corresponding geometric labels.

Inspired by Li et al. (2024b), we utilize LLMs to generate the textual description of the visual cues. Specifically, we design the prompts about subject and object to generate the accurate and rich descriptions of visual cues. The prompts we used can be found in appendix A.3. And then, we group them according to the category of objects. For the $i$-th object category, we denote the set of visual cues related to its categories as $TC_i \in \{TC_i^1, ..., TC_i^{m_i}\}$, where $m_i$ represents that there are $m_i$ cues related to object $i$.

Extensive research has shown that LLM contains important world knowledge. Thus, it can provide us with excellent guidance. However, we realize that these textual descriptions can not be directly used to infer visual relationships. In a visual scene, an object may interact with multiple other objects. These text descriptions are not fully applicable to the current object pair. Meanwhile, due to the factors such as occlusion, image contains the changes that are difficult to summarize in text. Thus, we need to combine the image content to filter the text.

A natural question that arises is: **how to effectively align text embedding and visual representation**? In this paper, we utilize large Vision Language Model(VLM) to address this problem. VLMs like CLIP(Radford et al. (2021)) are pretrained on web-scale datasets consisting of image-text pairs, resulting in a high degree of alignment between visual text. Many previous works have also proved its ability to capture fine-grained visual cues.

Specially, for the relationship proposal from object $i$ to $j$, we extract their clip patch embeddings $pe_{ij} \in R^{14 \times 14 \times 512}$ in their union box through the visual encoder of CLIP. And then, we input the textual descriptions of their visual cues $TC_i$ and $TC_j$ into the text encoder to obtain the corresponding text embeddings $te_{ij} \in \{te_{ij}^1, ..., te_{ij}^{(m_i+m_j)}\}$. Followed by, we treat text embeddings as a set of queries, and perform global similarity calculation and local cross-attention with patch embeddings, respectively.

**Global similarity calculation.** As shown in the green-background region in Fig.2. Given the relationship proposals from object $i$ to $j$, we utilize a convolutional network $f_g$ to extract their global representation $gv_{ij}$,

$$gv_{ij} = f_g(pe_{ij}) \tag{18}$$

and then we calculate the similarity $sim_{ij} \in \{sim_{ij}^1, ..., sim_{ij}^{(m_i+m_j)}\}$ between each text embedding and it. For the $k$-th text embedding $te_{ij}^k$, its similarity to $gv_{ij}$ is defined as $sim_i^k j$,

$$sim_{ij}^k = cos\_sim(gv_{ij}, te_{ij}^k) \tag{19}$$

where $cos\_sim$ is the cosine similarity method. Each similarity value represents the degree of correlation between the visual cue and the current object pair.

**Local cross attention.** As shown in blue-background region in Fig.2. Every text embedding are treated as a query to perform a convolution operation on patch embeddings $pe_{ij}$ and get the spatial activation map $sp_{ij} \in \{sp_{ij}^1, ..., sp_{ij}^{(m_i+m_j)}\}$,

$$sp_{ij}^k = sigmoid(pe_{ij} \odot te_{ij}^k) \tag{20}$$

where $\odot$ represents the convolution operation. Each value in the spatial active map represents how likely this local region contains the corresponding visual cues.

After the above convolutional operation, for each patch embedding we get s set of spatial activation maps corresponding to text embeddings. Then we utilize these spatial activation maps to perform region-based attention and weighted average pooling on original patch embeddings. We utilize the activation values as the the pooling weights. Therefore, we can get $N$ visual embeddings corresponding to $N$ text embeddings. For the $k$-th visual embedding $vc_{ij}^k$, it can be computed by,

$$vc_{ij}^k = gap(pe_{ij} \otimes sp_{ij}^k) \tag{21}$$

where $\otimes$ denotes elements-wise product, $gap$ is the global average pooling. Finally, their known cues $kc_{ij}$ can be computed by,

$$kc_{ij} = sim_{ij} \cdot vc_{ij} \tag{22}$$

**Summary.** So far, we have captured the known cues that each visual relationship relies on, and evaluated their performance and existence in the image. **It seems that the visual relationship can be inferred by directly matching them with the definition**. However, it requires a premise: we can ensure that the definition is complete and accurate and can cope with all image changes.

Obviously, it is difficult for us to complete this requirement. The image contains rich changes, and it is difficult for us to summarize them completely with text. But it does not mean that these textual descriptions are invalid. They can provide an excellent guide for us to infer visual relationships, which is also the core idea of our design of KCC strategy. We show some results of our KCC strategy in fig. 6.

A.4 UNKNOWN CUES EXTRACTION

Through the above operations, we first utilize LLMs to collect known cues. And then we measure their existence in the image through **global similarity calculation** strategy, and capture the visual representation of these cues through **local cross attention** strategy. However, we realize that these cues may not be complete.

Visual perception is a rich signal for modeling a vastness of experiences in the world that cannot be documented by text alone. In fact, it is difficult for us to fully summarize the cues on which a visual relationship depends. For example, although the "upright leg" is the key factor of our inference

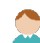

You are a teacher who teaches people about visual relationships. Please explain the following questions step by step.

Absolutely! Please go ahead and ask your questions, and I'll walk you through them step by step. 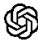

What are the useful visual features of subject and object that can distinguish the relation "**walking on**", when subject belongs to "**human**" and object belongs to "**location**"?

**Human features:**
1) The human body is typically upright, with a slight forward lean;
2) Arms often swing alternately with leg movement, indicating motion;
3) One leg is typically forward, while the other is behind;
....
**Location Features:**
1) Roads, sidewalks, paths are often level or have a gentle slope;
2) The surface is usually extended horizontally or at a slight incline, which allows for walking rather than climbing;
3) Walkable locations tend to have a solid texture (e.g., pavement, dirt path, grass), offering support to the human foot during walking;
...

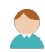

What are the useful visual features of subject and object that can distinguish the relation "**parked on**", when subject belongs to "**vehicle**" and object belongs to "**location**"?

**Vehicle features:**
1) The vehicle should be directly above or on the surface of the location (e.g., road, driveway, parking lot);
2) The bottom part of the vehicle (wheels or base) should touch or be very close to the surface of the location;
3) The vehicle should have a horizontal alignment consistent with being stationary. If it appears tilted, it might indicate movement, not parking;
....
**Location Features:**
1) The location should typically be flat (or mostly flat) and appear solid, such as asphalt, concrete, or gravel;
2) Locations like grassy fields, sand, or dirt might still allow parking, but it could look different;
...

Figure 5: The scene graph of visual scene and corresponding geometric labels.

of "stand on", the "upright body" may also be used as a basis for our judgment when the leg is occluded. Thus, text description can be used as an excellent guidance, but should not be used as the only criterion. We call these cues that are not/are difficult to describe in text as **unknown cues**. Our model must have the ability to extract unknown cues from images.

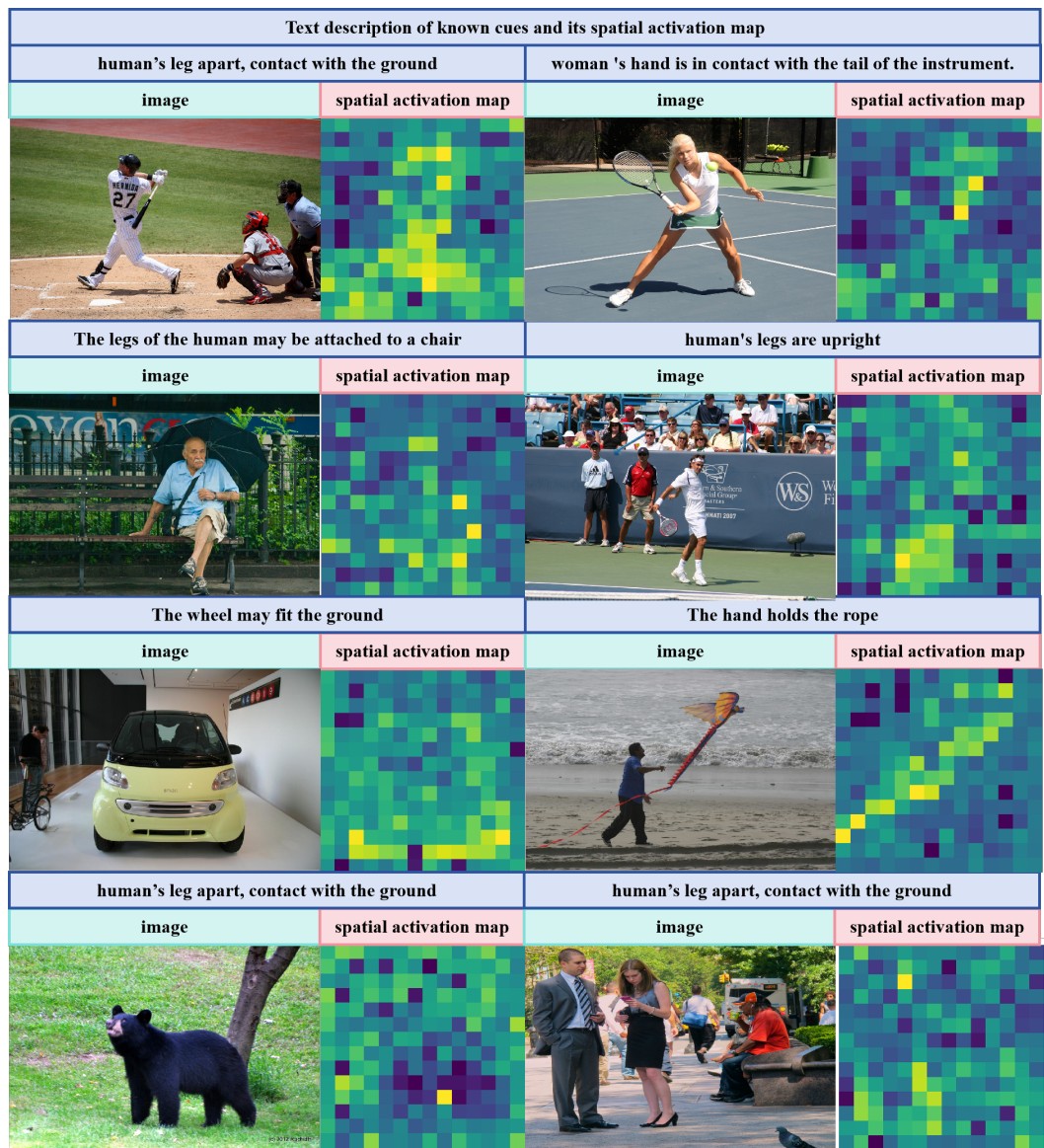

Figure 6: Some text descriptions of known cues and their spatial activation maps.

Since the visual relationship depends on the object, we believe that the information that helps to infer the visual relationship should be at least related to the object. Therefore, we perform a simple decoupling of the object. As shown in fig.2, we generates $P$ convolutional filters independently based on class semantic knowledge. Concretely, for the $i$-th object proposal, we get its class semantic knowledge $ck_i$ of its category through clip text encoder. And then, we design $P$ different MLPs to decouple it. Every MLP independently maps the class semantic vector from semantic space to a $1 \times 1$ convolutional filter in visual space. In the filed of few-shot learning, this decoupling method is simple and common. Obviously, there are many better studies on object decoupling, but it is not our focus.

Followed by, for the relationship proposal from object $i$ to $j$, we can get $2 \times P$ latent parts $lp$. And every latent part are treated as a query to perform a convolution operation on $pe_{ij}$ and get the spatial activation map of unknown cues $unsp_{ij} \in \{unsp_{ij}^1, ..., unsp_{ij}^{2 \times P}\}$,

$$unsp_{ij}^k = sigmoid(pe_{ij} + lp_{ij}^k) \qquad (23)$$

Each value in the spatial active map represents how likely this local region contains the corresponding visual cues.

After the above convolutional operation, for each patch embedding we get a set of spatial activation maps corresponding to latent parts. Then we utilize these spatial activation maps to perform region-based attention and weighted average pooling on original patch embeddings. We utilize the activation values as the pooling weights. Therefore, we can get $2 \times P$ unknown cues corresponding to $2 \times P$ latent parts. For the $k$-th unknown cues $uc_{ij}^k$, it can be computed by,

$$uc_{ij}^k = gap(pe_{ij} \otimes unsp_{ij}^k) \tag{24}$$

It is worth mentioning that **UCE** does not conflict with **KCC**. **UCE** is designed to allow the model to explore the visual cues related to objects on its own, so it is likely to search for cues consistent with **KCC**.

A.5    PROGRESSIVE INFERENCE

In general, our progressive inference strategy is based on geometric relationships and gradually optimizes predicate representation. It consists of two parts: for geometric relationship, its predicate representation follows the existing works, that is the initial predicate representation we mentioned in section 3.2; for non-geometric relationship, its predicate representation can be constructed by the following steps.

**Build hierarchy.** Firstly, we construct a hierarchy to reflect the similarity of predicates. Following the existing works(Zhang et al. (2024a); Wang et al. (2019)), we construct this hierarchy according to the semantic similarity of predicates. Specifically, according to the semantic embedding of predicates, we utilize **hierarchical clustering** strategy to build this hierarchy. The detailed description can be found in appendix A.2.

**Node representation calculation.** In this hierarchy, the nodes in the last layer are meaningful predicate labels. We can easily get their representations. However, for other nodes in the hierarchy, they have no practical significance and are only generated during our clustering process. Thus, we first calculate the representation of these nodes. Suppose that the $k$-th node of the $l$-th layer has $n$ child nodes in the $l + 1$-th layer. And its node representation $hn_l^k$ is the average value of its child node representations,

$$hn_l^k = \frac{1}{n} \sum_{j=1}^{n} hn_{l+1}^j \tag{25}$$

If the node has no subsequent nodes, it indicates that the node is a specific predicate, and the representation of this node can be calculated by the following steps. Firstly, we get the text embedding of all predicates $tp$ by clip text encoder. And then for the relationship proposal from object $i$ to $j$, we calculate the similarity $ps_{ij}$ between each predicate and the visual cues obtained by our **KCC** and **UCE** strategies,

$$ps_{ij}^k = cos\_sim((kc_{ij} \oplus uc_{ij}), tp^k) \tag{26}$$

Finally, the node representation for the last layer in our hierarchy $hn_{la}$ can be computed by,

$$hn_{la} = tp + ps_{ij} \cdot (kc_{ij} \oplus uc_{ij}) \tag{27}$$

It is worth mentioning that in the clustering process of our hierarchy, we utilize the word embedding from **glove**, while the computational node representation utilizes the text embedding from **clip**. The specific reasons can be found in our appendix A.2.

**Predicate representation optimization.** For the non-geometric predicates, their initial representations are the visual embedding from the clip visual encoder. We will optimize them according our hierarchy. Given the relationship proposal from object $i$ to $j$, we denote their predicate representation in $l$-th layer as $re_{ij}$. Followed by, we can calculate the similarity between it and all nodes in $l + 1$-th layer as $sh_{ij}^{l+1}$,

$$sh_{ij}^{l+1} = softmax(\frac{re_{ij}^l \cdot hn_{l+1}}{d_k}) \tag{28}$$

where $d_k$ is the dimension of these embeddings(in this work, its value is 512). Then, the predicate representation in $l+1$ can be computed by,

$$re_{ij}^{l+1} = re_{ij}^l + sh_{ij}^{l+1} \cdot hn_{l+1} \tag{29}$$

Finally, for the relationship proposal from object $i$ to $j$, their final predicate representation $r_{ij}$ can be computed by,

$$r_{ij} = f_r(rg_{ij} \oplus re_{ij}^{la}) \tag{30}$$

where $f_r$ is a fully connected network to unify the dimension and $re_{ij}^{la}$ is the last representation through the above optimization process.

### A.6 SUMMARY AND LIMITATIONS

In general, in order to solve the shortcomings of the existing method to represent predicates, we propose a novel method - **P**rogressive **V**isual **R**elationship **I**nference(**PVRI**) - which considers both rough visual appearance and fine-grained visual cues to gradually infer visual relationships. Obviously, we adopt an extremely simple strategy for decoupling of objects, and there are more advanced works in few shot learning that can improve our **UCE** strategy, but it is not our research focus. As an important part of scene understanding, the accuracy and interpretability of visual relationship are very important for us to understand the scene. We also look forward to more research attempts to better collect visual cues.

