# OpenReview forum: "Progressive Visual Relationship Inference"
_ICLR.cc/2025/Conference — ICLR 2025 Conference Withdrawn Submission_

### Official Review · Reviewer_RBLa · 2024-11-01

**Soundness:** 3
**Presentation:** 1
**Contribution:** 2
**Rating:** 3
**Confidence:** 4

**Summary:**

This paper propose a novel method - Progressive Visual Relationship Inference(PVRI) - which considers both rough visual appearance and fine-grained visual cues to gradually infer visual relationships. It conducted experiments on Visual Genome, OpenImages datasets. The experiments show the effectiveness of its method.

**Strengths:**

The authors designed ablation studies to verify the importance of the two strategies, KCC and UCE.

**Weaknesses:**

1. The experiments of this paper is not as substantial as other state-of-the-art papers in this field. There is a lack of comparison with the latest methods (e.g. HiKER-SGG: Hierarchical Knowledge Enhanced Robust Scene Graph Generation).
2. The writing of this paper is very poor. The whole article is messy and does not highlight the key points. Some of the descriptions are redundant, such as the experimental part (lines 470-485), which is obvious in the table 1. However, the description of the experimental background and the reasons for the experimental results is not enough. There are many similar problems.
3. The main innovations of this paper are concentrated in two points: one is the geometric space representation of relationship, and the other is the clues provided by LLM. However, these two innovations are not the latest, referring to Li's paper (Zero-shot Visual Relation Detection via Composite Visual Cues from Large Language Models).
4. There are few pictures in the paper, which are not intuitive. The pictures of the pipeline do not explain the method very well. Even the borders of some pictures(e.g. fig 1) are not processed. The author should re-work this part.

**Questions:**

The experiments lack comparisons with some of the latest state-of-the-art methods, such as HiKER-SGG. Why it happens?

---

### Official Review · Reviewer_GpQ9 · 2024-11-02

**Soundness:** 3
**Presentation:** 1
**Contribution:** 2
**Rating:** 3
**Confidence:** 4

**Summary:**

This paper proposes a new method named PVRI for scene graph generation, which integrates both rough visual appearance and fine-grained visual cues. It introduces Known Cue Collections (KCC), which uses Large Language Models to gather detailed descriptions, and Unknown Cue Extraction (UCE), which focuses on visual cues. The author conducts comprehensive experiments to demonstrate the effectiveness of these methods using the Open Images V6 and Visual Genome datasets.

**Strengths:**

1. The paper proposes a new method that complements the rough visual appearances by using a Large Language Model to extract the detailed descriptions of cues.
2. The author demonstrates the effectiveness of this proposed method through extensive experiments.
3. This is an innovative attempt to capture fine-grained features using the Large Language Model.

**Weaknesses:**

1. There is no analysis regarding complexity or cost at all. It seems that the author utilizes the costly GPT-4 to extract a detailed description of visual cues for *all triplets*. Therefore, I'm afraid of the significant costs involved in using the GPT-4 for the scene graph generation task, considering it deals with numerous triplets. Meanwhile, the author utilizes the Large Language Model, large vision-language model (i.e., CLIP), and BGNN [1], which are all heavy. In this regard, the complexity analysis is necessary.

2. The trade-off between mR@K and R@K appears to be more pronounced than in other baselines, leading us to question the effectiveness of the proposed model. Therefore, the author could use the harmonic average or simple average of R@K and mR@K, which are widely used for showing the trade-off in the nature of scene graph generation [2,3].

3. The author should include state-of-the-art baselines in the comparison. While there are several recent methods for scene graph generation, the author only compares performance with a few others. For example, the author evaluates head, body, and tail performance against BGNN, which is from 2022. To clearly validate the effectiveness of the proposed method, it is essential to incorporate state-of-the-art baselines like VETO [3] and HIKER-SGG [4]. Additionally, since the author discusses the problem of the hierarchical tree structure of HIKER-SGG, it makes sense to compare performance with that model as well.

4. There is no evidence supporting the effectiveness of the interpretability of the proposed method. Although the author mentions in the Introduction that existing works lack interpretability and attempts to address this issue, there is no analysis provided. This absence raises doubts about how the proposed method actually enhances interpretability.

[1] Bipartite Graph Network with Adaptive Message Passing for Unbiased Scene Graph Generation. Li et al. CVPR'21
[2] Fine-Grained Scene Graph Generation with Data Transfer. Zhang et al. ECCV'22
[3] Vision Relation Transformer for Unbiased Scene Graph Generation. Sudhakaran et al. ICCV'23
[4] Leveraging Predicate and Triplet Learning for Scene Graph Generation. Li et al. CVPR'24

**Questions:**

1. What would happen if GPT-4 were replaced with an open-source large language model or another model? I’m interested in whether the proposed method can be generalized to other large language models, which would enhance its robustness.

---

### Official Review · Reviewer_G21i · 2024-11-02

**Soundness:** 2
**Presentation:** 2
**Contribution:** 1
**Rating:** 3
**Confidence:** 5

**Summary:**

This paper proposes Progressive Visual Relationship Inference (PVRI), a method to improve visual relationship understanding by combining rough appearances with detailed visual cues. PVRI includes three steps: gathering known cues from language models, extracting unknown cues directly from images, and progressively refining relationship inferences. Tested on Visual Genome and Open Image V6 datasets, PVRI shows strong performance in distinguishing subtle visual relationships, enhancing both accuracy and interpretability in scene understanding.

**Strengths:**

1. The overall structure of the paper is complete.
2. The content of the paper is quite substantial.

**Weaknesses:**

1. Poor presentation. Line 93～97 are almost the same as abstract and line 161-192. Moreover, the interpretation of “Progressive Inference” is very confused and DO NOT reflect the concept of “progressive” at all.
2. Novelty is limited. There are many works[1][2][3] using LLM to generate visual cues to assist SGG model to distinguish similar relation categories. The author does not even discuss these efforts.
[1] LLM4SGG: Large language models for weakly supervised scene graph generation, CVPR 2024.
[2] Zero-shot visual relation detection via composite visual cues from large language models, NeurIPS 2023.
[3] Less is more: Toward zero-shot local scene graph generation via foundation models, ArXiv 2023.
3. Low performance. Both the recall@K and mean recall@K are lower than PE-Net-Reweight in Table 1 of [1].
[1] Prototype-based embedding network for scene graph generation, CVPR 2023.
4. Rough drawing. Fig 1 (b) is not a vector image, and the text cannot be seen clearly.
5. Insufficient experiments. Lack of ablation study of PI and the robustness of the multiple description generation.
6. Due to the lack of comparison between training and inference time. the calculation similarity of the union region requires N*N cropping (N is the number of the object in an image), which is very time-consuming.

**Questions:**

Please refer to the weaknesses.

---

### Note · Authors · 2025-01-05

I have read and agree with the venue's withdrawal policy on behalf of myself and my co-authors.